# Red Flags in Primary Mitochondrial Diseases: What Should We Recognize?

**DOI:** 10.3390/ijms242316746

**Published:** 2023-11-25

**Authors:** Federica Conti, Serena Di Martino, Filippo Drago, Claudio Bucolo, Vincenzo Micale, Vincenzo Montano, Gabriele Siciliano, Michelangelo Mancuso, Piervito Lopriore

**Affiliations:** 1Department of Biomedical and Biotechnological Science, School of Medicine, University of Catania, 95123 Catania, Italy; conti.federica1@hotmail.it (F.C.); serena.dm.92@gmail.com (S.D.M.); claudio.bucolo@unict.it (C.B.); vincenzomicale@inwind.it (V.M.); 2Center for Research in Ocular Pharmacology-CERFO, University of Catania, 95213 Catania, Italy; 3Neurological Institute, Department of Clinical and Experimental Medicine, University of Pisa, 56126 Pisa, Italypiervito.lopriore@gmail.com (P.L.)

**Keywords:** primary mitochondrial diseases, red flags, mitochondria, rare diseases

## Abstract

Primary mitochondrial diseases (PMDs) are complex group of metabolic disorders caused by genetically determined impairment of the mitochondrial oxidative phosphorylation (OXPHOS). The unique features of mitochondrial genetics and the pivotal role of mitochondria in cell biology explain the phenotypical heterogeneity of primary mitochondrial diseases and the resulting diagnostic challenges that follow. Some peculiar features (“red flags”) may indicate a primary mitochondrial disease, helping the physician to orient in this diagnostic maze. In this narrative review, we aimed to outline the features of the most common mitochondrial red flags offering a general overview on the topic that could help physicians to untangle mitochondrial medicine complexity.

## 1. Introduction

PMDs are the most frequent genetic metabolic disorders in humans, with a prevalence of approximately 1 in 4300 cases [1]. Pathogenic variants in all 37 mitochondrial DNA (mtDNA) genes and in more than 400 nuclear DNA genes have been associated with direct or indirect defects of the OXPHOS, which is the hallmark of these diseases [2]. These unique features of mitochondrial genetics and their ubiquitousness in human tissues explain PMDs’ vast clinical heterogeneity. PMDs are multisystemic diseases, mainly affecting high-energy demanding tissues, such as the muscles and central and peripheral nervous systems. Age of onset is variable, with the most severe defects generally appearing in childhood [3]. Cellular mtDNA polyplasmy, tissue heteroplasmy (proportion between normal and mutant mtDNA variants), mitotic segregation (mtDNA’s stochastic distribution during cell division), and the phenotypic threshold effect (tissue specific sensitivity to oxidative metabolism defects), are not sufficient to explain PMDs’ clinical heterogeneity [2,4,5]. For example, they cannot elucidate tissue specific alteration in individuals harboring homoplasmic mtDNA mutation, the absence of clear correlation between heteroplasmy level and disease severity, or the heterogeneous manifestations in nuclear DNA-related PMDs [4]. For mtDNA-related PMDs, environmental factors, other nuclear or mitochondrial mutations, mtDNA haplotype or tissue-specific nuclear gene expression patterns, and a complex molecular and metabolic cellular compensation response to OXPHOS defect may act as modifiers [2,4,5]. For nuclear DNA-related PMDs, the lack of complexity comprehension is even greater. It is important to underline that some of the abnormalities of nuclear genes involved affect mtDNA replication and the expression system (mtDNA maintenance), causing dysfunctions of the intergenomic signaling (manifesting with multiple mtDNA deletions or mtDNA depletion), and have combined features of mendelian and mitochondrial genetics [2]. Because of the vast clinical and genetic variability, diagnosing PMDs can be challenging. Given this, with this narrative review we aimed to dissect the most common “red flags” that could help clinicians in recognizing, diagnosing, and managing PMDs. For each red flag, we have outlined the pathogenetic mechanism, its presence in certain syndromic pictures, and the association with other phenotypic manifestations. For this purpose, to help the readers orient themselves in this maze, we summarized in Table 1 the main phenotypic characteristics of well-defined PMDs and their main genetic etiology.

## 2. Neurological Red Flags

### 2.1. Progressive External Ophthalmoplegia (PEO)

Chronic progressive external ophthalmoplegia PEO (cPEO) is a mitochondrial syndrome characterized by myopathy of extraocular muscles. Indeed, the extraocular muscles are dependent on oxidative metabolism and for this reason are highly vulnerable to mitochondrial dysfunction [4]. cPEO can initiate at any age with progressive bilateral eyelid ptosis and diffuse ophthalmoparesis. Typically, progressive ptosis represents the earliest and the most noticeable feature of PEO patients and leads to the gradual acquisition of a chin-up compensatory head position [6]. In addition, severe fatigue, depression, and pain might contribute to disease impact on the quality of life of PEO patients [7]. Moreover, dysphagia and decreased respiratory muscle strength, as well as obstructive apneas, can be present [8].

Indeed, the term “pure PEO” identifies patients with isolated ocular myopathy, while when PEO occurs with other symptoms of mitochondrial dysfunction with neuromuscular involvement, it is denoted as “PEO plus syndrome”. The symptoms more frequently associated with “PEO plus syndrome” comprise muscle weakness (43%), exercise intolerance (23%), muscle wasting (18%), hearing loss (15%), and swallowing impairment (15%). Furthermore, Orsucci and colleagues proposed the term “PEO-encephalomyopathy” to indicate patients with PEO combined with signs of central nervous system (CNS) pathology [9]. PEO can be even a clinical sign of a multisystem disease; indeed, it is one of the most frequent phenotypes of PMDs [9]. Particularly, PEO is a traditional clinical manifestation of Kearns–Sayre syndrome (KSS), together with pigmentary retinopathy and Pearson syndrome, together with sideroblastic anemia [10].

A retrospective study on a large cohort from the database of the “Nation-wide Italian Collaborative Network of Mitochondrial Diseases” showed that ocular myopathy was more often associated with mtDNA single large-scale deletions and *POLG1* mutations, while it was less common in patients with *OPA1* mutations and all the mtDNA point mutations, especially in tRNAs. Moreover, “PEO-encephalomyopathy” was often associated with the m.3243A>G mutation. Importantly, patients with *POLG1* mutations seem to have more severe quality of life impairment compared to patients with mtDNA mutations or deletions [9].

### 2.2. Exercise Intolerance

Exercise intolerance is a feature which can alert clinicians to mitochondrial myopathies, though it is difficult to be clinically assessed. Indeed, it is an elusive symptom, which sometimes could be missed. However, exercise intolerance can be quantified through measurement of the maximal oxygen uptake, by evaluating resting and peak exercise-induced serum lactate levels, as well as by examining lactate/pyruvate ratios [11]. Owing to exercise performance limitations, patients with exercise intolerance can be mistaken for having cardiopulmonary diseases.

According to the Italian Network of Mitochondrial Disorders database, more than 20% of mitochondrial patients complain of exercise intolerance [12], which seems to be associated with specific mutations, as well as m.3243A>G. About one-third of patients with exercise intolerance symptoms showed increased creatine kinase (CK) levels. In addition, ragged red fibers (RRFs) and cytochrome c oxidase (COX)-negative fibers were often present in subjects with exercise intolerance [12].

### 2.3. Stroke-like Episodes (SLEs)

Stroke-like episodes (SLEs) are the defining features of mitochondrial encephalopathy with lactic acidosis and stroke-like episodes (MELAS) syndrome, a PMD often associated to multisystem involvement, such as diabetes, lactic acidosis, and hearing loss [10].

SLEs are characterized by headache, nausea and vomiting, encephalopathy, focal-onset seizures, or psychiatric disorders. In addition, epilepsia partialis continua and, sometimes, status epilepticus might even occur during the stroke-like episode. Due to its high energy demand, the brain is particularly vulnerable to mitochondrial oxidative phosphorylation defects [13]. A SLE, which can have an acute or subacute neurological manifestation, typically affects individuals younger than 40 years and it is caused by focal brain dysfunction of an epileptic nature. Late-onset SLEs have been rarely reported. [14].

About 80% of the cases are associated with the pathogenic m.3243A>G mutation in the *MT-TL1* gene, the first genetic defect linked to MELAS syndrome [15]. However, other rare mtDNA mutations and recessive *POLG1* mutations are well-recognized causes of SLEs [16,17]. Specifically, one-third of patients with the m.3243A>G mutation manifest SLEs after 40 year of age [18], whereas SLEs are more aggressive in patients harboring *POLG1* mutations and appear earlier. Indeed, *POLG1* patients have an explosive onset and a rapid progression, with a higher risk of death due to status epilepticus [17], in comparison to patients with 3243A>G mutation, who mainly die subsequently to complications not related to CNS. In particular, 3243A>G patients show a protracted disease course with an insidious onset [19,20]. Moreover, around half of *POLG1* patients have no other symptoms outside of CNS features before the first SLE. On the contrary, m.3243A>G patients manifest preceding systemic features, which can be useful as “red flags” to forecast SLEs in PMDs associated with m.3243A>G mutation [20]. On brain magnetic resonance imaging (MRI) scans, SLEs show involvement of the cortex and juxtacortical white matter and typically are not confined to vascular territories. SLEs are characterized by an increased lactate level, both in the lesion area and in the apparently unaffected brain regions, evidenced by MRI proton spectroscopy, which can facilitate the screening and early diagnosis [3,21]. The neurological disability severity depends on the level of temporal, parietal, and/or occipital lobe involvement, evidenced through neuroimaging. Focal seizure activity can be detected by means of electroencephalography. Recurrent SLEs, usually followed by symptomatic complete or incomplete remissions, lead to progressive accumulation of neurological deficits.

### 2.4. Bilateral Brainstem Lesions

Bilateral brainstem lesions represent the typical neuropathological features of Leigh syndrome (LS), together with lesions of the cerebellum, diencephalon, and/or basal ganglia, which lead to a progressive decline in neurological function [10]. Indeed, bilateral symmetrical lesions extend from the basal ganglia and thalamus, through brainstem structures, to spinal cord posterior columns [22]. It should be noted that neurons become affected by an energy supply deficit because of their high energy demand [23].

LS is the most common presentation of pediatric mitochondrial disease and a progressive neurodegenerative disorder, affecting about 1 in 36,000 births [10,24]. Most affected patients die before 3 years of age, usually from sudden respiratory failure; indeed, the outcome of LS is poor [25]. The clinical hallmarks of LS include psychomotor delay or regression, hypotonia, tremor, weakness, truncal ataxia, and lactic acidosis [26], as well as dystonia [27]. There are even extra-CNS features in LS, as well as polyneuropathy, myopathy, diabetes mellitus, short stature, hypertrichosis, cardiomyopathy, anemia, sleep disturbances, renal failure, gastrointestinal dysfunction, hearing loss, failure to thrive, retinitis pigmentosa, cranial nerve palsies, and scoliosis. Leigh-like syndrome refers to patients with atypical symptoms, laboratory findings, and radiologic features, but with the overall clinical picture indicative of LS. In particular, Leigh-like patients manifest late-onset myopathy, and few or no ocular motility abnormalities, while a respiratory deficit is usually not present [28].

LS is an example of the genetic heterogeneity of PMDs. It is caused by mutations in about 100 genes across both nuclear and mitochondrial genomes [10]. Indeed, LS is commonly associated with mitochondrial complex IV (cytochrome c oxidase-COX) deficiency, which can result from mutations in several COX structural proteins or assembly factors, such as *SURF1* [29], which represents the main cause of COX-deficient LS. Pathogenic variants in mtDNA-encoded complex I subunits or *MT-ATP6* are strongly associated with subacute brainstem dysfunction in young patients [30,31]. Brainstem dysfunction has been evidenced even in patients with neuropathy and ataxia (NARP) associated with *MT-ATP6* pathogenic variants, without a clear diagnosis of mitochondrial inherited LS (MILS) [30]. Noteworthy, La Morgia and colleagues reported an association between a rare variant (m.4171C>A) in *MT-ND1*, causing Leber’s hereditary optic neuropathy (LHON) and bilateral brainstem lesions together with vomiting and vertigo, in addition to the typical feature of LHON [32]. In addition, progressive brainstem lesions have been found even in patients with KSS [33,34].

At the microscopic level, neuronal loss, proportionate loss of myelin, reactive astrocytosis, and proliferation of cerebral microvessels are present [4]. A brain MRI with proton spectroscopy sequences is useful for diagnosis of LS. The affected regions appear symmetrical and hyperintense on T2-weighted sequences. Furthermore, increased lactate levels in the blood or cerebrospinal fluid represents another important clue for the diagnosis of LS [35].

### 2.5. Epilepsy

Epilepsy is one of the most common CNS features of PMDs. The pathophysiology of epilepsy in PMDs is not clearly understood, even if seizures can be explained mainly by an impaired oxidative phosphorylation leading to ATP deficiency [36].

Seizures are hallmarks in some PMDs, such as Alpers–Huttenlocher syndrome (AHS) [17], pyruvate dehydrogenase complex deficiency (PDHc) [37], LS [38], myoclonic epilepsy with RRFs (MERRF) [39], and MELAS syndrome [40]. The incidence rate of epileptic seizures among adult PMDs patients is about 10–40% and is 60% in pediatric patients [41]. PMD patients often experience myoclonus and various types of focal seizures, or even tonic seizures, tonic–clonic seizures, and infantile spasms. Moreover, patients manifest status epilepticus, including nonconvulsive status epilepticus and epilepsia partialis continua, which is a focal motor status epilepticus occurring for a minimum of one hour and recurring at ten seconds intervals [36,42]. Interestingly, status epilepticus is more common in mitochondrial depletion syndromes (MDS) or mtDNA-related PMDs, and usually coexists with SLEs [36]. Of note, epilepsy in MELAS is heterogeneous; however, focal onset, clonic, and tonic seizures represent the most common SLEs-associated epileptic phenomena [43].

It is noteworthy that repeated or long-lasting seizures prolong the energy deficit, leading to further mitochondrial dysfunction, which conducts to subsequent seizures. Indeed, this self-perpetuating cycle may lead to the development of SLEs, especially in MELAS and *POLG*-related diseases, often related to status epilepticus.

Mitochondrial epilepsy is usually challenging to treat, often leading to progressive epileptic encephalopathy and a worse prognosis [36]. In particular, epilepsy due to the mutation in the *POLG1* gene is refractory to pharmacotherapy [44]. Regarding treatments, epilepsy associated with PMDs can be managed as non-mitochondrial epilepsy [43]. Indeed, levetiracetam in association with benzodiazepines is the first line therapy for myoclonus in MERFF [36], whereas lamotrigine might promote myoclonic seizures [45]. Patients with MELAS or drug-resistant seizures can be treated with zonisamide and lacosamide [36,43]. Additionally, midazolam, along with alternative interventions, as well as perampanel, corticosteroids, ketamine, immunoglobulin, and magnesium infusion are employed for the treatment of status epilepticus [43,46,47]. To obtain seizure control, most patients need to be treated with multiple anti-epileptic drugs and therapy must be personalized for each patient. Nevertheless, it is important to not use drugs with mitochondrial toxicity, such as sodium valproate, especially in patients harboring *POLG1* mutations in order, to avoid seizure aggravation and hepatic failure [48]. Currently, new therapies focused on antioxidant approaches, mitophagy and mitochondrial biogenesis, or stabilization of the mitochondrial membrane are under investigation [49].

Importantly, there is no typical electroencephalography trace for seizures in PMDs, even if background activity is usually disturbed. Regarding neuroimaging, the analysis of 1467 patients with PMDs revealed that MRI abnormalities were significantly more common in subjects with epilepsy compared to patients without seizures [50]. Indeed, brain atrophy, stroke-like lesions, basal ganglia, and white matter changes are more common in PMDs patients with epilepsy [51].

### 2.6. Movement Disorders

Movement disorders in PMDs are not rare and, even when isolated, these should incite a search for PMD diagnosis. In particular, ataxia and parkinsonism represent the most common movement disorders [52]. Muscle and nerve cells have high energy needs and ATP derived from mitochondria represents their main source of energy [53]. Indeed, neurological and muscular symptoms are common features of PMDs. Cerebellar ataxia, one of the most prevalent neurological mitochondrial signs [52], is characterized by movement incoordination or imbalance and leads to abnormal movement and gait, caused by dysfunction of the cerebellum or its connections [54]. Cerebellar ataxia is often associated with abnormal eye movements, dysmetria, kinetic tremor, dysarthria, and/or dysphagia, other symptoms of cerebellum dysfunction [10]. However, sensory ataxia is characterized by an alteration in the proprioceptive system due to spinal or peripheral lesions. Stepping gait, which aggravates with loss of visual fixation, is a feature of sensory ataxia [52]. Cerebellar ataxia can be the onset symptom of well-defined mitochondrial syndromes, such as KSS, MILS, MERRF, and can also be one of the clinical symptoms in undefined mitochondrial encephalopathies [55]. In addition, isolated ataxia may even be due to autosomal recessive gene defects, including *POLG*-related diseases, such as mitochondrial recessive ataxia syndrome (MIRAS) [56]. Autosomal recessive coenzyme Q10 deficiency, a treatable PMD, is characterized by isolated cerebellar ataxia. In addition, ataxia can be associated with ocular myopathy and with psychiatric comorbidities or, rarely, epilepsy [57], such as in infantile-onset spinocerebellar ataxia (IOSCA), a neurodegenerative disorder caused by *TWNK* mutations [10,58]. Sensory ataxia is a less common feature among PMDs and mainly characterizes neuropathy, ataxia, and pigmentary retinopathy syndrome (NARP) and sensory ataxia neuropathy dysarthria and ophthalmoplegia (SANDO) [10,59]. Recently, recessive mutations in the *PTRM1* gene, encoding for a mitochondrial matrix enzyme involved in mitochondrial proteostasis, were found to be associated with complex early-onset phenotypes including spinocerebellar ataxia [60]. Neuroimaging may be supportive for the diagnosis; indeed, a brain MRI can show peculiar alterations depending on the syndrome, together with cerebellar atrophy [52].

Mitochondrial parkinsonism is usually difficult to diagnose due to similarities with idiopathic Parkinson’s disease. Usually, the disease onset is at around fifty years of age [10]. Both nuclear gene mutations and mtDNA genetic abnormalities can cause mitochondrial parkinsonism. Between nuclear genes, the most prevalent are *POLG1* and *TWNK* [54]. These genes should be tested, especially in families with autosomal dominant PEO and parkinsonism. Levodopa-responsive parkinsonism has been reported in patients with either biallelic or heterozygous *POLG1* mutations [61]. Among mtDNA genetic abnormalities, the m.8344A>G mutation in the *MT-TK* gene, causing MERRF syndrome, was described in one patient with levodopa-responsive parkinsonism [62]. Furthermore, *MT-ND1, MT-ND6* gene mutations and a 4-base-pair deletion in the cytochrome B gene were found in patients with parkinsonism associated with optic atrophy, myoclonic epilepsy and SLEs, typical PMDs red flags [63,64,65]. To conclude, the diagnosis of mitochondrial parkinsonism should have to be accounted for in the presence of other signs of mitochondrial dysfunction, as well as PEO, neuropathy, myopathy, ataxia, isolated muscle pain, psychiatric disorders, and epilepsy.

Noteworthily, along with ataxia and parkinsonism, in a large cohort study, Montano and colleagues reported other movement disorders, such as tremor and dystonia, as clinical features of adult-onset PMDs [66].

### 2.7. Migraine

Migraine represents another significant neurological red flag of PMDs [10]. Mitochondrial dysfunction plays a central role in the pathophysiology of migraine, which reflects the vulnerability of CNS to respiratory chain defects. However, clear pathomechanisms are not delineated [67]. Importantly, due to the absence of a structured diagnostic headache tool, specific questionnaires, interviews, and objective clinical examination, the prevalence of migraine in PMDs patients fluctuates noticeably in the scientific literature [68]. A cross-sectional cohort study reported that migraine has a prevalence of 35% in PMDs patients, which is higher compared to the general population. This prevalence was reported to be independent from sex, phenotype, or genotype [68]. Moreover, the same study observed that epilepsy, myoclonus, and SLEs are more frequent in patients with migraine compared to patients without it. Noticeably, migraine might be an early manifestation of the disease [68]. Specifically, MELAS patients harboring m.3243A> G point mutation can suffer from repeated episodes of migraine, which is one of the most frequent symptoms [69,70]. Noteworthily, pharmacological management of PMDs patients with migraine could be particularly challenging, probably due to poor tolerance to drugs commonly used for headaches [71].

### 2.8. Neuropathy

Peripheral neuropathies occur in a third of patients with PMDs. Peripheral nerves are highly dependent on energy metabolism, as well as mitochondria that play a pivotal role in maintaining viability of motor and sensory neurons and their axons [72]. Neuropathies are particularly associated with defects in mitochondrial DNA maintenance and replication, as well as defects in the respiratory chain, such as complex V. However, the number of genes and phenotypes linked with PMDs and neuropathies is rapidly increasing [72].

Peripheral neuropathy can manifest as a minor clinical sign in the majority of PMDs, with a small impact on patients’ lives. However, moderate to severe peripheral neuropathy might be the first manifestation in specific syndromes, such as KSS, LHON plus, mitochondrial neurogastrointestinal encephalomyopathy (MNGIE), MERRF, MELAS, and LS [72,73,74,75,76,77,78]. Peripheral neuropathy associated with PMDs mainly manifests as chronic sensorimotor axonal polyneuropathy [79]. Specifically, sensory ataxic neuropathy, presenting in isolation or in combination with other signs/symptoms, is usually associated with NARP and *POLG*-related disorders (especially SANDO phenotype).

Mutations in the *MFN2* and *GDAP1* genes encoding proteins involved in mitochondrial dynamics have been evidenced as causes of Charcot–Marie–Tooth neuropathy (CMT) [80]. CMT neuropathy represents the most common inherited neurological disorder, affecting around 1 in every 2500 people in the world [81,82]. The typical CMT phenotype comprises distal weakness, sensory loss, foot deformities, as well as absent ankle reflex, manifesting in most patients in the first or second decade of life. However, many patients develop severe disability in infancy or early childhood and there are patients which manifest few or no symptoms of neuropathy until adulthood [82]. The majority of CMT neuropathies are demyelinating, while up to one-third are the result of a primary axonal disorder [83,84]. Mutations in *MFN2* and *GDAP1* affect axonal mitochondrial transport and function and have been associated with axonal forms of CMT. The mitochondrial dynamics alterations affect peripheral axons from spinal cord motor and dorsal root ganglia neurons, causing peripheral neuropathies [85].

Noteworthily, muscle biopsy, neuroimaging, cerebrospinal fluid analysis, and metabolic and genetic tests in blood and muscle are essential for the diagnosis of peripheral neuropathies [82].

### 2.9. Cognitive Decline

Cognitive decline may represent a clinical manifestation of PMDs, even if it remains one of the least understood aspects [86]. How the cognitive state might change with PMDs’ progression remains unclear. Most of the studies reporting cognitive impairment in PMDs patients are mainly retrospective analyses of patients with high genotypic variability and additional risk factors [87]. According to El-Hattab and colleagues < 10% of MELAS patients showed developmental delay as am initial clinical manifestation, while 10–24% of MELAS patients manifested impaired mentation during the disease course. In addition, 50–74% of MELAS patients revealed learning disability or memory impairment, while ≥90% of patients experienced dementia [40]. However, generalizing these findings to a wider population of PMDs patients is challenging [88,89]. More recently, disease severity has been demonstrated to be a stronger predictor of cognitive abilities compared to genotype [87]. Importantly, Kaufmann and colleagues [90] found a progression of general cognitive difficulties over a 4-year period in fully symptomatic MELAS patients with the m.3243A>G genotype. In addition to MELAS, cognitive impairment may be a clinical feature of other PMDs, such as MERRF syndrome [91], LS [92], NARP [93], and LHON [94]. Cognitive decline is also a prominent feature in *PTRM1*-related phenotypes. Interestingly, *PITRM1* mutations have been found to cause cerebral amyloid beta accumulation, significantly linking mitochondrial functioning with amyloidotic neurodegeneration [60]. Noteworthily, brain MRI findings varied considerably between those studies, leading to a weak association between cognitive impairments and neuroimaging results [87].

### 2.10. Optic Atrophy and Pigmented Retinopathy

Common clinical features of PMDs include ocular manifestations, such as optic atrophy and retinopathy. Mitochondria are essential for the survival and function of retinal ganglion cells (RGCs), one of the neuronal populations of the retina. Given their strong dependency on oxidative metabolism, RGCs are particularly susceptible to mitochondrial deficit [95].

LHON is the most common inherited optic nerve disease, hallmarked by subacute bilateral loss of central vision due to dysfunction and loss of RGCs [96]. LHON primarily affects healthy young males compared to females [97]. Indeed, gender is a risk factor, partially explainable by the protective effect of estrogens [98]. Environmental factor, such as smoking, play a role in LHON pathogenesis [99]. The onset of visual loss in LHON is usually unilateral, with the second eye becoming affected within weeks or months [3]. Optic disc hyperemia, oedema of the peripapillary retinal nerve fiber layer, and retinal telangiectasia represent typical ophthalmological features in the acute phase of LHON [100]. Furthermore, thickening of the retinal nerve fiber layer has been described before angiopathic changes [101] LHON is caused by mtDNA point mutations, with the m.11778G>A in *MT-ND4* being the most common and the one associated with poorer outcomes [102]. Other common LHON mutations associated with LHON disease include m.3460G>A in *MT-ND1* and m.14484T>C in *MT-ND6* mutations [103,104]. Thus, the vast majority of LHON are accounted by three mtDNA mutations encoding for mitochondrial complex I subunits.

Autosomal dominant optic atrophy (ADOA) is another PMD affecting RGCs. As with LHON, ADOA is characterized by central vision deterioration caused by optic nerve atrophy and retinal nerve fiber layer degeneration [105]. ADOA is associated with mutations in the nuclear-encoded gene *OPA1*, which is involved in several cellular processes, including the mitochondrial dynamics [10]. Indeed, around 50–60% of ADOA cases are caused by *OPA1* mutations, with a prevalence of 1 in 25,000 people [100]. Blindness occurs in 85% of ADOA patients, manifesting at a mean age of 8 years [10]. As with LHON, in about 20% of cases, ADOA patients can manifest extra-ocular neurological phenotypes (ADOA-plus) [4]. The most frequent “plus” manifestation is bilateral sensorineural deafness, described in late childhood and early adulthood; other clinical manifestations, such as ataxia, peripheral neuropathy, and ocular and skeletal myopathy might appear later, in the third decade of life. These patients’ muscle biopsies show multiple mtDNA deletions and COX-deficient fibers, given the *OPA1* role in mt-DNA maintenance [106]. The proportion of COX-negative fibers is much higher in patients with ADOA-plus compared to non-syndromic ADOA patients [4].

Pigmentary retinopathy, resulting from retinal degeneration, may occur in PMDs patients. It is characterized by progressive photoreceptor damage and subsequent cell death, leading to the atrophy of photoreceptors and adjacent retina layers [107,108]. mtDNA single large-scale deletions are associated with diffuse pigmentary retinopathy [109]. Indeed, pigmentary retinopathy, together with PEO, is a primary clinical feature of KSS [10], whereas mtDNA point mutations are mainly related to clinically more localized macular pigmentary and atrophic changes [107]. Therefore, patients harboring m.3243A>G mutation may manifest pigmentary retinopathy as the most common ocular sign [110]. During ophthalmoscopy investigation, pigmentary retinopathy is defined by the presence of retinal pigment epithelium, hyperpigmented areas close to the back of the eye. However, pigmentary retinopathy can be even asymptomatic, remaining undiagnosed. Finally, pigmentary retinopathy can also occur in patients with LS or cPEO, as well as in patients with maternally inherited diabetes and deafness syndrome (MIDD) [111,112,113].

### 2.11. Sensorineural Hearing Loss

Hearing impairment is a common clinical feature of PMDs, both as an isolated sign and as part of a multisystem phenotype. Usually, hearing loss in PMDs manifests at a young age after speech acquisition (post-lingual) with other systemic clinical signs of PMDs, as well as with a maternal pattern of inheritance [114,115]. Sensorineural hearing loss is mainly caused by degeneration and loss of hair cells in the cochlea of the inner ear, which are highly dependent on mitochondrial metabolism [116,117].

The mitochondrial non-syndromic hearing loss (NSHL) is an example of isolated manifestation of hearing loss, which is associated with several mtDNA variants, especially in *MT-RNR1* or *MT-TS1* [118]. Moreover, sensorineural hearing loss can be part of a mitochondrial syndrome spectrum, such as MELAS, KSS, MERFF, and MIDD [10]. Indeed, sensorineural hearing loss may be a clinical feature in patients with *POLG1* and *SPG7* mutations [114]. In addition, hearing loss represents the most prevalent extra-ocular clinical feature in ADOA associated with the *OPA1* gene [115].

Due to their prokaryotic origins, mitochondria are vulnerable to antibiotics which target the bacterial ribosome, particularly in presence of specific mtDNA mutations. Specifically, patients harboring m.1555A>G or m.1494C>T in *MT-RNR1* may manifest deafness after exposure to aminoglycoside antibiotics. Indeed, the screening test for mtDNA mutations could be useful before starting the treatment [119].

## 3. Gastrointestinal Red Flags

### 3.1. Liver Failure

Acute or fulminant liver failure is a significant indicator of PMDs. In particular, mitochondrial liver disease manifests primarily in children, either in isolation or association with CNS involvement [120]. Mitochondria represent the major energy source for liver cells; indeed, impaired mitochondrial functions may contribute to liver damage [121]. Recessive mutations in the nuclear-encoded gene *TRMU*, encoding for a tRNA-modifying protein, may lead to acute liver failure in the first months of life [122]. 

Liver disease associated with chronic neuromuscular disease or other multisystem involvement might be a clinical sign of PMDs. Liver failure is a clinical feature of AHS, a severe pediatric encephalopathy, associated with recessive *POLG1* mutations [123]. AHS is hallmarked by liver degeneration and encephalopathy characterized by intractable seizures, with frequent episodes of epilepsia partialis continua or status epilepticus, and by progressive developmental regression. Liver failure may precede or occur after seizure onset. Terminal liver impairment is common [124]. Sodium valproate treatment should be avoided in AHS because it might precipitate liver failure [119,125]. Other MDS, such as recessive *TWNK* mutations, can be associated with signs of liver impairment, together with severe early onset encephalopathy, describing a phenotype reminiscent of AHS. In particular, clinical features include hypotonia, athetosis, sensory neuropathy, ataxia, hearing deficit, ophthalmoplegia, intractable epilepsy, and elevation of serum transaminases. Liver biopsy shows mtDNA depletion, whereas the muscle mtDNA can be only slightly affected [126]. Mutations in *DGUOK* and *MPV17*, two genes encoding mitochondrial proteins involved in mtDNA maintenance, cause infantile hepatocerebral MDSs [127]. Mutations in *DGUOK,* encoding for the mitochondrial deoxyguanosine kinase, specifically causes MDS type 3, while *MPV17* is associated with MDS type 6 [128]. MDS type 3 is characterized by early-onset severe liver disease, low-birth weight, and neurological impairment, such as myopathy and nystagmus, whereas MDS type 6 is characterized by infantile or childhood onset progressive hepatic disease and systemic manifestations, such as hypoglycemia and lactic acidosis. In both cases, liver dysfunction progress to chronic liver failure, the most common cause of death in these PMDs [129]. According to some investigations the levels of tyrosine and phenylamine are elevated in the blood or urine of newborns harboring *DGUOK* and *MPV17* mutations [130,131]. Mutations in *ETHE1* can also cause liver failure. *ETHE1* encodes for a mitochondrial protein that plays a key role in hydrogen sulfide detoxification as a sulfur dioxygenase. *ETHE1* mutations lead to the toxic accumulation of H_2_S and metabolites in tissues and body fluids, causing a severe fatal metabolic disorder known as ethylmalonic encephalopathy, a complex disorder characterized by, in addition to liver dysfunction, chronic diarrhea, petechial rush, and neurological manifestations [132,133]. Liver transplantation could increase the survival of patients with *DGUOK* and *ETHE1* mutations [134,135].

### 3.2. Severe Dysmotility/Pseudo-Obstructive Episodes

Gastrointestinal dysmotility, due to degeneration of gastrointestinal tract muscles, is a red flag for PMDs, usually encountered with multiple genetic defects [3]. The pathophysiology underling this phenomenon remains unclear [136]. Progressive gastrointestinal dysmotility is a severe clinical feature of MNGIE, in combination with neuromuscular involvement (PEO, polyneuropathy, hearing loss, and leukoencephalopathy) [137]. Gastrointestinal signs/symptoms comprise diarrhea, abdominal pain, vomiting, pseudo-obstruction which may lead to weight loss, and cachexia. 

MNGIE is an autosomal recessive syndrome caused by mutations in the *TYMP* gene leading to thymidine phosphorylase deficiency. It usually manifests at a mean age of 18 years and leads to death in early adulthood. MNGIE is frequently misdiagnosed; according to Hirano and colleagues, there is limited evidence regarding diagnostic and therapeutic approaches due to the lack of controlled studies with proper follow-up [138]. Interestingly, an observational cohort study reported that severe gastrointestinal pseudo-obstruction is much more common in PMDs patients with the pathogenic m.3243A>G mtDNA mutation, despite the fact gastrointestinal symptoms are usually under-recognized in PMDs. For example, pseudo-obstructive episodes are common in MELAS patients, especially during the acute phase of a SLE. Considering the frequency of m.3243A>G mtDNA mutation, clinicians should always assess the gastrointestinal system in those patients to mitigate the risk of developing severe intestinal pseudo-obstructions [139]. 

A clinical phenotype close to MNGIE syndrome has been reported in five patients with *POLG1* mutations, who had prominent symptoms of persistent diarrhea and cachexia related to gastrointestinal dysmotility, as well as ptosis, proximal myopathy, and sensory neuropathy [140,141,142]. Finally, gastrointestinal dysmotility, feeding difficulties, and failure to thrive are gastrointestinal manifestations observed in the infantile hepatocerebral MDSs due to *MPV17* mutations [143].

## 4. Cardiovascular Red Flags

### 4.1. Cardiomyopathies

The heart is one of the most metabolically active tissues in the human body, relying heavily on mitochondrial ATP production to maintain the high energy demand of individually contracting cardiac myocytes. Adults with PMDs are indeed about 30% more likely to have abnormal electrocardiograms and/or echocardiograms. 

During cardiac MRI, more than one-third of these patients exhibit late gadolinium enhancement linked to myocardial fibrosis. Cardiomyopathies are the most frequent cardiac manifestations of PMDs and are estimated to occur in 20–40% of patients [144]. Hypertrophic cardiomyopathies are most common, but dilated, restrictive, and other types are also seen. The severity ranges from asymptomatic, sometimes spontaneously reversible conditions to a severe cardiomyopathy with an early, even prenatal, onset that causes death in early infancy. The presence of cardiomyopathy in a PMD, regardless of its severity, is associated with a poorer prognosis [145]. Cardiomyopathies have been reported in patients with MELAS in 30–32% of cases [70] and LS in 18–21% of cases. In LS patients with cardiac manifestations, hypertrophic cardiomyopathy was found in 50% of cases [146]. 

Two syndromic cardiomyopathies presenting in infancy are Barth syndrome and Sengers syndrome. Barth syndrome is an X-linked condition characterized by dilated cardiomyopathy, skeletal myopathy (mostly proximal), poor growth, (cyclical) neutropenia, and 3-methylglutaconic aciduria [147]. Hypertrophic cardiomyopathy, left ventricular noncompaction, and endocardial fibroelastosis are further cardiac characteristics. Barth syndrome is caused by mutations in the *TAZ* gene, which alter the inner mitochondrial membrane’s ability to produce cardiolipin [148]. 

Mutations in the gene encoding for acylglycerol kinase (*AGK*), which is also a part of the mitochondrial import machinery, cause Sengers syndrome, a rare autosomal recessive PMD [149,150]. Hypertrophic cardiomyopathy and congenital cataract are present in affected newborns. Lactic acidosis, exercise intolerance, and skeletal myopathy are additional clinical characteristics. Infant mortality from progressive heart failure is possible, but long-term survival has been observed in some patients. 

Additionally, other causes of childhood onset mitochondrial cardiomyopathies are increasingly recognized, including disorders of mitochondrial translation and defects of coenzyme Q10 biosynthesis [151].

### 4.2. Arrythmias, Cardiac Conduction Defects, and Valvulopathies

Arrythmias, conduction defects, and pulmonary hypertension are examples of other rarer cardiac manifestations [152]. Prolonged QT and arrhythmias are common in patients with cPEO [153]. Cardiac conduction defects, often culminating in a complete heart block, have been reported frequently in patients with KSS. Biallelic variants in *PPA2* gene, encoding for a mitochondrial located pyrophosphatase, have been associated with sudden death [154]. According to a recent systematic review, patients with MELAS have significantly more electrocardiography and echocardiography alterations compared to patients with other PMD phenotypes [155]. A well-described cardiac complication in MELAS patients is represented by Wolff–Parkinson–White syndrome [155] a conduction abnormality characterized by the presence of altered electrically conductive circuits between the atria and ventricles. Cardiac conduction defects may also be present in MERRF, which is caused by m.8344A>G mutations in MT-TK in 80–90% of cases [156,157]. One case series found cardiac abnormalities in 53% of MERRF patients harboring this mutation [158]. The m.8363G>A mutation has also been reported in a case series of nine patients with clinic presentations fitting MERRF, and four of the nine patients were found to have cardiac conduction abnormalities [159]. 

Despite being a rare manifestation of PMDs, valvular heart disease has been linked to *PDSS1* mutations, which encodes for an enzyme that produces coenzyme Q10 [160]. Recently, it has been suggested that children with LS elicited by *ADAR* mutations are susceptible to developing fatal heart valve calcification [161].

## 5. Kidney Red Flags

### 5.1. Tubulopathies

The kidney primarily depends on aerobic metabolism. Due to its inability to synthesize ATP anaerobically, the cortical tubule, especially in its proximal part, is extremely susceptible to OXPHOS [162]. Numerous nuclear and mtDNA gene mutations have been linked to renal manifestations in PMDs. The most frequent condition is tubular dysfunction, which can range from a mild hyperaminoaciduria triggered by catabolic circumstances (i.e., sepsis, fever) to a full-blown de Toni–Debré–Fanconi syndrome. Depending on the location of tubule damage and the type of chemicals lost in the urine, the tubulopathy may have Fanconi- or Gitelman-like characteristics. Gitelman syndrome affects the distal tubule and causes more specialized losses, whereas Fanconi syndrome affects the proximal tubule and causes broader losses (i.e., glucosuria, phosphaturia, generalized aminoaciduria, and bicarbonate wasting). Both diseases can produce hypomagnesemia, but Gitelman syndrome is more commonly associated with it [163]. The most severe tubulopathies are typically associated with large-scale mtDNA deletions and KSS or Person syndrome phenotypes [164]. Moreover, MDSs (especially *RRM2B* defect but also *DGUOK*, *MPV17*, *SUCLA2*, and *TK2* mutations) and disorders of mitochondrial translation (such as *TFSM* gene mutations), are also associated with tubulopathy [165].

### 5.2. Glomerulopathies

Focal segmental glomerulosclerosis and steroid-resistant nephrotic syndrome have been reported in PMDs patients with the m.3243G>A mutation [166,167]. Focal segmental glomerulosclerosis is frequently observed in adults but rarely in children [167]. A pseudohypoaldosteronism-like picture with hyperkaliemia, hyponatremia, and gradual renal impairment can occur in children with *RMND1* mutations [168]. Some defects in the CoQ biosynthesis pathway (*PDSS2*, *COQ2*, *COQ6*, *COQ8* deficiencies) are also associated with glomerular disease and may respond to treatment with CoQ10 [169].

## 6. Endocrine Red Flags

### 6.1. Diabetes Mellitus

Because steroid hormones are synthesized in the mitochondria, reduced ATP production affects hormone production and results in endocrinological symptoms. Overall, it appears that abnormalities brought on by mtDNA flaws, particularly large-scale deletions and point mutations in tRNA genes, are more likely to have endocrinological symptoms. These symptoms may also be present in patients with nuclear gene abnormalities, which most typically involve mtDNA maintenance and translation-related problems [170]. The endocrine manifestation that is best characterized is diabetes mellitus. The impairment of the mitochondria’s function as a glucose sensor, which links glucose metabolism to insulin release, contributes to the mechanism of diabetes in PMDs in addition to decreased insulin secretion brought on by a lack of ATP supply [171]. Diabetes is reported in a substantial proportion of patients carrying the mutation m.3243A>G in *MT-TL1*, either as a dominant feature in MIDD, in multisystem phenotypes, or as a part of MELAS. The m.3243A>G mutation is estimated to cause 0.5–2.9% of diabetes mellitus in the population [170]. Diabetes mellitus is also frequently seen in KSS or in Pearson syndrome, where exocrine pancreas dysfunction is a more prominent feature [172].

### 6.2. Other Endocrine Signs

Additional endocrinological manifestations that should be mentioned are hypothyroidism, hypoparathyroidism, adrenal insufficiency, and hypogonadism [170,173,174]. Premature ovarian failure is observed in *POLG*-related diseases and in association with leukoencephalopathy in *AARS2* deficiency. Although isolated endocrine involvement may be the first sign of a mitochondrial problem, mitochondrial endocrine dysfunction most usually arises in the context of multisystem disease [170].

## 7. Other Red Flags

### 7.1. Lactic Acidosis

Lactic acidosis is the most well-known laboratory finding in patients with PMDs. ATP synthesis is reduced because of malfunction in the electron transport chain. Low ATP levels activate glycolysis, which overproduces pyruvate. Extra pyruvate is either converted to lactate or transaminated to alanine. Congenital lactic acidosis has most frequently been reported in children with genetically determined complex I deficiency [175]. However, lactic acidosis has historically being recognized as a feature of different PMDs. Lactic acidosis may be transient and resolve within a few days of birth in some PMDs [176]. A total of 50% of 51 PMDs patients with different ages investigated by Jackson and colleagues had lactic acidosis [177]. Patients with mtDNA abnormalities are more likely to develop lactic acidosis [178], and mtDNA mutations are responsible for 10% of all PMDs in children [179]. In a study performed by Hutchesson and colleagues in 11 infants with known PMD, increased plasma levels of lactate were found [180]. Munnich and colleagues [178] carried out the biggest investigation of lactic acidosis, specifically in childhood-onset PMDs. A higher venous lactate level was detected in 30% of the 235 children investigated. 

Although common in children, severe lactic acidosis is rare in adults with PMDs. In adults, lactic acidosis is frequently observed associated with other mitochondrial manifestations, such as stroke-like episodes, encephalopathy, seizures, and myopathies [181]. Lactic acidosis is a low-specificity biomarker: it can be caused by, among others, venous stasis, hypoxia, hypoperfusion, hepatic dysfunction, renal failure, drug toxicity, sepsis, spasticity, hyperinsulinism, chronic thiamine deficiency, and seizures [178,180].

### 7.2. Cachexia

Cachexia is a wasting illness elicited by mitochondrial disfunctions, so occurs in PMDs, especially associated with MELAS or MNGIE [137]. Cachexia is a syndrome that, by definition, affects skeletal muscle mass and function with or without body fat loss [182]. Furthermore, cardiac dysfunction brought on by striated muscle atrophy, as well as respiratory distress due to its effects on respiratory muscle function, are both primary causes of death [4]. Numerous inflammatory cytokines are involved in the etiology of cachexia, including tumor necrosis factor alpha, interleukins 1 and 6, and interferon gamma [182]. Systemic inflammation is one of the major underlying mechanisms driving the reduction in skeletal muscle mass [183]. In rodents, early impairment of mitochondria quality control and function has been reported [183,184] and has been observed to trigger events occurring before muscle atrophy in cachexia-related muscle atrophy [185]. In animal models of cachexia, the enhanced activity of the ubiquitin–proteasome pathway may drive muscle wasting [183,186]. In humans, activation of this system is not a consistent finding, suggesting the activation of other pathways, such as autophagy [187] and apoptosis [188].

### 7.3. Lipomas

Rarely, PMDs may present with multiple systemic lipomatosis (MSL) a rare disorder involving the adipose tissues and characterized by the development of non-encapsulated lipomas usually distributed in the cervical, cranial, and thoracic region. MSL has been reported in patients harboring point mutation in the mtDNA-encoded tRNA-lysine gene (*MT-TK*), which is the most frequent mutation associated with MERRF [189]. Muscle biopsies from roughly 28% of MSL patients revealed mitochondrial impairments, such as RRFs and COX-negative fibers [190]. The m.8344 A>G mutation in *MT-TK* has been identified in about 16% of MSL patients [191,192]. Additionally, MSL has recently been identified in a family with mtDNA multiple deletions harboring a mutation in the *MFN2* gene [193]. Furthermore, MSL was associated with PEO and cerebellar ataxia, but not with diabetes or dyslipidemia, in a cohort of Italian PMDs patients. Additionally, mitochondrial MSL is more common in women and those who have never abused alcohol, is frequently linked to muscle involvement, and can be asymmetrical. A mitochondrial dysfunction mechanism is clearly supported by the correlation between MSL and high lactate serum level.

### 7.4. Short Stature

A major physiological endpoint is the growth and multiplication of chondrocytes in the growth plate, which is under the influence of many stimuli during childhood and adolescence. The development process may be compromised by mitochondrial malfunction. Children with PMDs are roughly shorter than their counterparts who are not afflicted, according to Wolny and colleagues [194], and their body mass index (BMI) is also reduced as a result [195]. In a Chinese cohort of pediatric patients with the m.3243A>G mutation in *MT-TL1*, 73% of the children were found to have short stature [196]. Short stature and poor growth are common in different PMDs, especially MERRF, MELAS [10], NARP [197], Pearson syndrome, and KSS [10]. Intriguingly, patients with short stature are more likely to have diabetes and cardiovascular involvement, whereas patients with low BMI are associated with neurological symptoms, such as seizures, encephalopathy, and SLEs [139]. Short stature and low BMI are also linked to gastrointestinal problems, as well as hearing loss. 

Moreover, numerous cases of growth-hormone insufficiency in patients with PMDs caused by m.3243A>G, single large-scale mtDNA deletions, and nuclear-encoded gene mutations have been reported [198,199]. One of the proposed theories is that mitochondrial alterations can affect pituitary function [200]. However, other factors may contribute to short stature and low BMI, such as lower oral intake due to poor appetite and early satiety linked to gastrointestinal dysmotility, gastroparesis, delayed stomach emptying, bacterial overgrowth, intestinal pseudo-obstruction, and dysphagia [201]. PMDs patients may develop malabsorption, and mtDNA abnormalities have been demonstrated to disrupt the function of the human colonic epithelium [202]. 

Patients with PMDs could also experience additional comorbid conditions, which could hypothetically have an impact on bone health and growth [170,203]. Endocrinopathies, including hypoparathyroidism, hypothyroidism, and diabetes, are among them. Growing evidence suggests that growth may be impacted by systemic inflammation and stress in chronic illness states [204]. Hypothyroidism is rare in adults with PMDs, and mitochondrial diabetes typically appears beyond the age of 35, well past the period of childhood and adolescence when growth occurs [205].

## 8. Combinations of Symptoms

Certain constellations of symptoms may also be clues to specific PMDs. For example, the combination of ataxia and myoclonus, traditionally linked to the progressive myoclonus–ataxia syndrome and progressive myoclonus epilepsy spectrum, may represent a red flag for MERRF syndrome. The combination of ataxia and myoclonus with other manifestations, such as SLEs, hearing loss, diabetes, optic atrophy, and cognitive impairments, may be related to MELAS [10]. Another example is parkinsonism, which may suggest a mitochondrial etiology in the case of association with optic atrophy or PEO [206]. The combination of diabetes mellitus and sensorineural hearing loss can be the manifestation of different PMDs, especially if diagnosed at young age in patients with low or normal BMI [207]. For example, the maternal heritability of diabetes or impaired glucose tolerance in addition to hearing impairment and maculopathy may be a red flag for MIDD, which is typically caused by m.3243A>G [208]. However, the association of diabetes and deafness is also observed in other non-mitochondrial syndromes, such as Wolfram syndrome (diabetes insipidus, diabetes mellitus, optic atrophy, and deafness (DIDMOAD)) [209], Rogers syndrome (megaloblastic anemia, diabetes mellitus, and sensorineural deafness) [210], or Herrmann syndrome (photomyoclonus, diabetes mellitus, deafness, and nephropathy) [211].

Finally, the combination of hepatopathy and encephalopathy occurring in patients with acute or chronic liver failure is related to MDSs. Affected infants typically present with growth failure, feeding difficulty, developmental delay, and hypotonia. Although brain involvement is usually prominent, in rare cases, *DGUOK* mutations may cause isolated liver failure. In *POLG*-related diseases with hepatopathy, brain involvement may not be immediately apparent, but developmental regression due to epileptic encephalopathy is inevitable [212].

## 9. Focus on Pediatric Mitochondrial Neurological Red Flags

The neurological features or symptoms present in the majority of children affected by PMDs are muscle weakness, peripheral neuropathy, ophthalmoplegia, movement disorder, epilepsy, and migraine, which illustrate why the neuropediatrician is one of the key physicians involved in diagnosis and management of childhood-onset PMDs. 

Seizures are a frequent complication of pediatric PMDs. PMDs patients may also present with epileptic syndromes, such as West and Lennox–Gastaut syndrome [213]. AHS, due to recessive mutations in *POLG1*, is the most common pediatric PMDs associated with epilepsy [214]. AHS patients often show focal, myoclonic, or complex seizures. Status epilepticus is common, sometimes starting with epilepsia partialis continua, followed by a generalized, therapy-resistant status. 

Gross and/or fine motor delay are also frequent complications in pediatric patients and may be linked to hypotonia or myopathy. Myopathy is often part of encephalomyopathy but can also occur as a pure manifestation. Usually, the term “floppy infant” is used to describe a newborn with poor muscle tone and weakness [215]. 

In the pediatric population, among movement disorders, dystonias are the most frequent symptoms, and are particularly seen in LS [27]. This is not surprising, as the syndrome includes lesions in the basal ganglia and other extrapyramidal structures, from which these types of symptoms arise. 

Another nonspecific CNS symptoms is developmental delay [216], which is usually global and affects cognitive, language, and motor skills. Given the enormous heterogeneity of PMDs, there is likely no unique cognitive profile. A study by Turconi and colleagues found that the non-verbal domain, notably the visuo-spatial ability, was more severely impaired. Working memory issues with verbal short-term memory were also noted [217]. Additionally, there are signs of autism spectrum disorders, and it has been suggested that mitochondrial malfunction may contribute to the illness process in autism spectrum disorders in general [218]. Generally speaking, psychiatric manifestations are common but often under-diagnosed in PMDs, including generalized anxiety, depression, and obsessive–compulsive spectrum disorder [219].

## 10. Discussion and Conclusions

The aim of this review was to describe the complexity of PMDs’ presentations and the existence of some peculiar features (“red flags”) that could help physicians on the diagnostic pathway; these are illustrated in Figure 1.

The peculiar symbiotic and semiautonomous nature of mitochondrial biology gives rise to a wide range of PMDs. For a genetic point of view, some typical phenotypes have been associated with specific genetic alterations. However, genotype–phenotype correlations, even for clinically defined syndromes, are complex and insufficient because of the genetic pleiotropy and the vast number of genes, either mitochondrial or nuclear-encoded, that may cause similar phenotypes. Moreover, while many patients fit into well-defined classic phenotypes, significant numbers are oligosymptomatic or have overlapping phenotypes. Generally, PMDs should be considered in any complex multisystem disorder, especially when neurological, eyes/ears, or endocrine manifestations are predominant. It is traditionally suggested that PMDs should be suspected in patients with an apparently unrelated involvement of two or more tissues.

If a PMD is suspected, family history is the first point to address; it must be taken meticulously, with special attention to minimal and apparently unspecific (“soft”) signs in the family. After the clinical assessment, neuroimaging, nerve conduction studies, electromyography, cardiac and retinal evaluation, as well as routine laboratory investigation, are important to support the diagnosis process and direct targeted molecular testing. In addition to phenotypic “red flags”, laboratory biomarkers are helpful for clinicians who suspect a PMD (Table 2). In recent years, specific guidelines for the diagnosis, management, and treatment of PMDs in pediatric patients and adults have been proposed [48,220,221]. The price of this diagnostic challenge is repaid by the large number of preclinical and clinical trials that have been developed, thanks to the implementation of national and international clinical registries, natural history studies and preclinical models. To date, therapies have been largely limited to supportive and symptomatic drugs, including vitamins and cofactors. However, small molecules approaches are emerging. They are principally focused on normalizing bioenergetic defects through different targets: modulation of oxidative stress, augmentation of mitochondrial biogenesis, regulation of autophagy, nucleotides pools restoration, restoration of the cellular NAD^+^ to NADH ratio, and others [222]. After idebenone for acute visual loss in LHON, only omaveloxolone for Friedreich’s ataxia have provided sufficient evidence for approval by U.S. Food and Drug administration [223]. Other promising therapeutics are allogenic stem-cell transplants, orthotopic liver transplantation, and enzyme replacement therapy in MNGIE, or deoxynucleoside supplementation therapy for TK2-deficient myopathy [138,224]. Gene therapy is another innovative approach. Allotopic AAV2-*ND4* gene therapy for LHON patients harboring the m.11778G>A mtDNA mutation is one example [225]. In experimental models, significant progress has been achieved in the development of anti-sense peptide nucleic acids, endonucleases, transcription activator-like effector nuclease (TALENS), zinc finger nucleases, and CRISPR-Cas 9 that traverse the mitochondrial membrane and directly target mutant mtDNA for degradation [226].

In conclusion, to the non-specialist, PMDs are complicated and puzzling, difficult to recognize, and challenging to diagnose. However, important advances have been made in the field of PMDs diagnostics and phenotyping, as well as in the generation of pre-clinical models and therapeutics. Given the exciting era that mitochondrial medicine is experiencing, and also thanks to international community efforts, there is more need to rase awareness and stimulate interest towards these disorders.

## Figures and Tables

**Figure 1 ijms-24-16746-f001:**
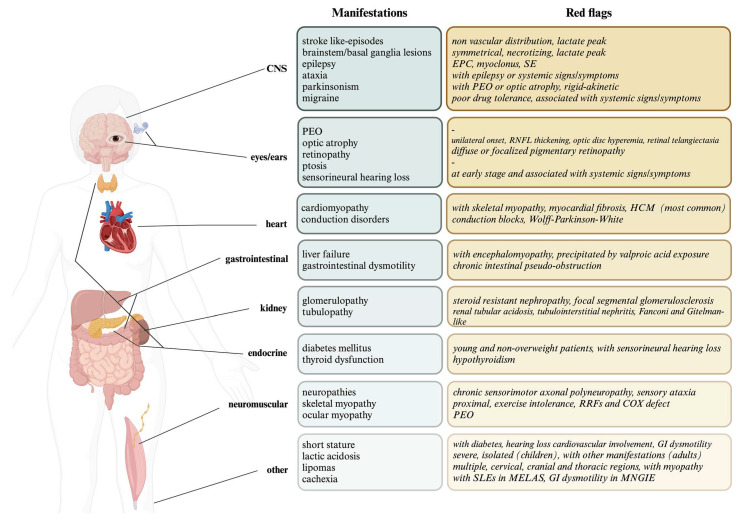
Graphical illustration of the main mitochondrial manifestations and red flags divided by system/organs.

**Table 1 ijms-24-16746-t001:** Key phenotypic features, red flags, and main genetic etiology of some examples of PMDs discussed in this review.

PMD	Key Phenotype Features	Most Common Genotype
Well-defined syndromes		
cPEO	Bilateral ptosis, progressive external ophthalmoplegia, mild exercise intolerance	mtDNA single large-scale deletion, nuclear genes involved in mtDNA maintenance (e.g., *POLG1*) (autosomal cPEO associated with multiple mtDNA deletions)
KSS	PEO (onset age < 20 years), pigmentary retinopathy, CSF protein >1 g/L, cerebellar ataxia, heart block	mtDNA single large-scale deletion
Pearson syndrome	Sideroblastic anemia of childhood, pancytopenia, exocrine pancreatic failure	mtDNA single large-scale deletion
MELAS	SLEs, epilepsy, lactic acidosis, dementia, diabetes mellitus, hearing loss, cardiomyopathy	m.3243A>G (*MT-TL1*)
Leigh syndrome	Infantile onset, bilateral brainstem lesions with subacute relapsing encephalopathy, cerebellar and brain stem signs	*MT-ATP6* >100 nuclear-encoded genes (e.g., *SURF1*)
NARP	Peripheral neuropathy, sensitive ataxia, pigmentary retinopathy	*MT-ATP6*
MERRF	Myoclonus, epilepsy, cerebellar ataxia, myopathy	m.8344A>G (*MT-TK*)
LHON	Subacute painless bilateral visual failure	m.11778G>A (*MT-ND4*), m.3460G>A (*MT-ND1*), m.14484T>C (*MT-ND6*)
ADOA	Subacute painless bilateral visual failure	*OPA1*
MIDD	Hearing loss, diabetes mellitus	m.3243A>G (*MT-TL1*)
MSL	Multiple lipomas, myopathy	m.8344A>G (*MT-TK*), *MFN2*
PMM	Proximal myopathy with or without PEO, exercise intolerance	mtDNA tRNAs (e.g., m.3243A>G in *MT-TL1*), mtDNA single large-scale deletion, *POLG1*, *TWNK*
Specific groups of disorders associated with multiple mtDNA deletions or mtDNA depletion		
ANS	MIRAS: mitochondrial recessive ataxia syndrome SANDO: sensory ataxia, neuropathy, dysarthria, ophthalmoplegia	*POLG1*
MEMSA	Epilepsy, myopathy, and ataxia without ophthalmoplegia	*POLG1*
AHS	Encephalopathy with intractable epilepsy, neuropathy, hepatic failure	*POLG1*
MCHS	Developmental delay or dementia, lactic acidosis, myopathy with failure to thrive, hepatic failure	*POLG1*
MNGIE	Severe gastrointestinal dysmotility, cachexia, PEO, sensorimotor neuropathy	*TYMP*
MDS	Myopathic forms: hypotonia, myopathy, feeding difficulty (“floppy infant”) Encephalomyopathic forms: hypotonia, global developmental delay Hepatocerebral forms: hepatic failure, developmental delay, abnormal eye movements, peripheral neuropathy (e.g., MCHS, *POLG*-related) Neurogastrointestinal forms: MNGIE	Mitochondrial nucleotide synthesis (*TK2, SUCLA2, SUCLG1, RRM2B, DGUOK, TYMP, MPV17*) mtDNA replication (*POLG1, TWNK*)

cPEO: chronic progressive external pphthalmoplegia; KSS: Kearns–Sayre syndrome; CSF: cerebrospinal fluid; MELAS: mitochondrial encephalomyopathy with lactic acidosis and stroke-like episodes; SLEs: stroke-like episodes; NARP: neurogenic weakness with ataxia and retinitis pigmentosa; MERRF: myoclonic epilepsy with ragged-red fibers; LHON: Leber hereditary optic neuropathy; ADOA: autosomal dominant optic atrophy; MIDD: mitochondrial inherited diabetes and deafness; MSL: multiple systemic lipomatosis; PMM: primary mitochondrial myopathy; ANS: ataxia neuropathy spectrum; AHS: Alpers–Huttenlocher syndrome; MCHS: Childhood myocerebrohepatopathy spectrum; MNGIE: mitochondrial neurogastrointestinal encephalomyopathy; MDS: mitochondrial DNA depletion syndromes. Please note that MEMSA includes the syndrome previously described as spinocerebellar ataxia with epilepsy (SCAE).

**Table 2 ijms-24-16746-t002:** Primary mitochondrial diseases biomarkers.

Biomarker	Description	Samples	SP, SE, AUC	Reference
Lactate	Quantification of lactate, which results from anaerobic ATP production	Blood, CSF, urine	SE: 34–62% SP: 83–100% [220] AUC: 0.8–0.926 [227,228]	[229]
Pyruvate	Quantification of pyruvate, which is a precursor of lactate	Blood, CSF, urine	SE: 63.6–83.3% SP: 87.2% [230] AUC: 0.907 [227] *L/P ratio* SP: 100% SE: 44% [231]	[232]
Creatine kinase	Quantification of creatine kinase, an enzyme released from damaged muscle fibers	Blood, CSF, urine	AUC: 0.609 [228]	[233]
Acylcarnitine	Quantification of acylcarnitine, which transports activated long-chain fatty acids to mitochondria, required for beta oxidation	Blood	-	[220]
Amino acids	Quantification of amino acids, as well as alanine, glycine, proline, and threonine, synthetized following respiratory chain dysfunction and altered cellular redox state	Blood, CSF, urine	-	[234]
GDF-15	Quantification of GDF-15, a member of the transforming growth factor beta family, induced in response to cellular stress	Blood	AUC: 0.70–0.999 [235,236] SE: 77.8% [237]	[238]
FGF-21	Quantification of FGF-21, a growth factor which regulates lipid and glucose metabolism, correlated with mitochondrial dysfunction	Blood	AUC: 0.69–0.9 [230,239] SE: 41.7–95.7% SP: 91.7% [230]	[239]
RRFs	Histological identification of RRFs, which represent the accumulation of abnormal mitochondria below the plasma membrane of the muscle fiber, linked with mitochondrial dysfunction (modified Gomori trichrome staining)	Muscle biopsy	-	[240]
RBFs	Histological identification of RBFs, which represent muscle fibers with increased levels of succinate dehydrogenase activity in the subsarcolemmal region	Muscle biopsy	-	[241]
COX-negative fibers	Histochemical identification of COX-deficient fibers, which represent a marker of mitochondrial respiratory chain deficiency	Muscle biopsy	-	[241]

SP: specificity; SE: sensitivity, AUC: area under curve; CSF: cerebrospinal fluid, RRFs: ragged red fibers, RBFs: ragged blue fibers, COX: cytochrome c oxidase. Note that the reported SP, SE, and AUC values refer to blood.

## Data Availability

Not applicable.

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
