# Peer review of "Red Flags in Primary Mitochondrial Diseases: What Should We Recognize?"

_ijms, 2023, doi:10.3390/ijms242316746_

Round 1

Reviewer 1 Report

Comments and Suggestions for Authors

The manuscript by Federica Conti and collaborators is a very well written, extensively documented and exhaustive review of the features of the most common “red flags “ of Primary Mitochondrial Diseases (PMDs).

This fantastic and necessary work represents a useful guide that could help physician to orient in the complex diagnosis of PMDs. It is a clear narrative review which is interesting to read with a good fit for IJMS. In general, I do have only minor points that should be addressed before the manuscript is ready for publication.

Line 25: PMDs are the most frequent GENETIC (or INEHRITED) metabolic disorders.

Line 129: …phosporilation defects [12] A DOT IS MISSED.

Line 142: According to recent data, LS affect 1 in 36.000 births.

Line 218: PMDS should be PMDs

SECTION 2.6. Spinocerebellar ataxia is also a prominent RED FLAG of neurodegenerative disorder associated with PITRM1 mutations.

SECTION 2.9:  Cognitive decline is one of the RED FLAG of neurodegenerative disorder associated with  PITRM1 mutations.

3.1: In this section, mutation in DGUOK and ETHE1 should be mentioned as both cause liver failure.

Author Response

Dear Reviewer,
thank you very much for your comments regarding our work. We have revised the manuscript according to your suggestions, in particular:
- we corrected the typos and did the minor revision reported (previous line 25, 129, 142, 218);
- we mentioned in sections 2.6 and 2.9 PITRM1 mutations;
- we expanded the section 3.1 describing DGUOK and ETHE1 mutations as  causes of liver failure.

Please note that all changes are highlighted in red. We all hope you will find the revised manuscript acceptable for publication.

Thank you very much and best regards

Michelangelo Mancuso

Reviewer 2 Report

Comments and Suggestions for Authors

The authors have prepared a great well-written review article disclosing the clinical hallmarks useful to navigate when examining a patient suspected to struggle with a mitochondrial disorder. This manuscript may be tremendously demanded by clinicists. However, given that the journal is rather focused on the molecular aspects of pathologies I would highly recommend to append some information in order to expand the audience:

-       A section with a brief explanation of the mechanisms underlying heterogenity of the clinical picture in MDs.

-       A small section/table with summarized description of biomarkers, helpful to detect MDs like creatine kinase levels, RRFs and COX-negative fibers (maybe rank them from more to less conventional/routine) 

-       A small section summarizing current challenges and reflecting recent advances in the treatment of MDs (including Gene Therapy)

Author Response

Dear Reviewer,
thank you very much for your work and valuable advice, that have enabled us to greatly improve the quality of our paper. We have revised the manuscript according to your suggestions, in particular:
- we added a small section regarding the mechanisms underlying phenotypic heterogeneity of PMDs (introduction paragraph);
- we added a table (table 2 in discussion paragraph) summarizing laboratory biomarkers helpful to recognize and diagnose PMDs; 
- we integrate the discussion paragraph with a small section regarding novel therapeutic approaches, recent advances and current challenges, giving a more complete conclusion to the entire manuscript.

Please note that all changes are highlighted in red. We all hope you will find the revised manuscript acceptable for publication.

Thank you very much and best regards

Michelangelo Mancuso